# Structure–Activity Relationship Target Prediction Studies of Clindamycin Derivatives with Broad-Spectrum Bacteriostatic Antibacterial Properties

**DOI:** 10.3390/molecules28217357

**Published:** 2023-10-31

**Authors:** Yiduo Jia, Yinmeng Zhang, Hong Zhu

**Affiliations:** School of Chemical Engineering and Pharmacy, Wuhan Institute of Technology, Wuhan 430073, China; jiayiduo0402@163.com (Y.J.); 17871722490@163.com (Y.Z.)

**Keywords:** clindamycin derivatives, MD simulation, binding force analysis, ADMET prediction, protein subcellular localization

## Abstract

This study investigated the potential of clindamycin derivatives with broad-spectrum antibacterial properties. The main goal was to identify new antibacterial targets to lay the foundation for developing novel antimicrobial agents. This research used molecular docking and dynamics simulations to explore how clindamycin derivatives could combat bacterial resistance and widen their antibacterial capabilities. Three different clindamycin derivatives were studied against 300 target proteins. Among these, 26 proteins were found to be common targets for all three derivatives. After further screening through molecular docking and dynamics simulations, four specific protein targets were identified. Notably, one of these targets, cell division protein FtsZ, was found to be primarily located in the cyto and cyto_nucl compartments. These findings suggest that clindamycin derivatives have the potential to address bacterial resistance and broaden their antibacterial effectiveness through these identified protein targets.

## 1. Introduction

Antimicrobial resistance (AMR) is an alarming global crisis, an issue of such magnitude that the World Health Organization (WHO) has designated it as one of the top 10 global public health threats facing humanity. The grim reality persists: drug-resistant infections exacted an unprecedented toll, contributing to a staggering 4.95 million deaths worldwide in 2019. Without decisive intervention, the trajectory of AMR is ominous, with projections indicating that global deaths attributable to AMR could skyrocket to an astonishing 10 million annually by the year 2050 [1]. Consequently, there is a pressing need to identify more efficacious treatments for drug-resistant infections [2]. To achieve this goal, the discovery and validation of novel therapeutic targets is of critical significance. It is essential to uncover fresh mechanisms for inhibiting bacterial growth to mitigate the risk of extensive resistance dissemination, thereby safeguarding the effectiveness of newly developed antimicrobial agents. For instance, clavulanic acid is an antibiotic that enhances the bactericidal effect of other antibiotics by inhibiting β-lactamase. However, it is only effective against specific types of β-lactamase and not others, thus limiting its spectrum of antibacterial activity. On the other hand, ciprofloxacin is an antibiotic that causes bacterial DNA damage and cell death by inhibiting DNA topoisomerase. However, misuse of ciprofloxacin has led to bacteria developing resistance mutations to quinolones in response to the prolonged stimulation of a single target, thereby increasing the risk of bacterial resistance. These findings underscore the importance of drug research, especially in the quest for broader-spectrum and more durable antibiotics to address the ever-growing challenge of antibiotic resistance. Antibacterial susceptibility primarily manifests in three aspects, firstly, Afzal, M et al. discovered that the RND-type efflux systems are a key contributor to resistance [3]. Inhibitors of these systems significantly reduce the antimicrobial concentrations required to combat resistant strains. Secondly, Rajasekhar, S. et al. found that hetero-antibiotics can overcome resistance in *Pseudomonas aeruginosa* by enhancing outer membrane permeability and reducing efflux [4]. Clindamycin is the preferred treatment for Gram-positive bacteria, with a strong inhibitory activity against *Staphylococcus aureus* (65 nM < MIC < 125 nM). However, its effectiveness against other Gram-negative bacteria and fungi is limited. In our previous research, clindamycin derivatives showed promise, demonstrating improved antimicrobial activity and broader spectrum coverage. The antimicrobial spectrum is outlined in Table 1 (the specific results are shown in the Appendix A). These three classes of clindamycin derivatives exhibit notable antimicrobial activity (32.5 nM < MIC < 125 nM) against six categories of bacteria, including four Gram-negative and two Gram-positive bacteria and one fungus. However, the specific mechanisms underlying their antimicrobial action warrant further investigation.

Targeted precision drug design is a pharmaceutical development strategy aimed at creating personalized medications tailored to specific diseases or disease subtypes based on individual patients’ biological characteristics and specific molecular targets. This approach centers on the discovery and validation of specific protein targets within bacteria, which play crucial roles in the survival and reproduction processes of these microorganisms. Once these targets are identified, researchers can design drug molecules with precision to disrupt the function of these targets [2,5]. 

For example, Streptococcus pneumoniae, a common bacterium, is responsible for various infections, including otitis media, pharyngitis, and pneumonia [4]. Notably, Gram-positive strains of this bacterium produce beta-lactamase, leading to resistance against beta-lactam antibiotics. Hence, the beta-lactamase of the Gram-positive strains of Streptococcus pneumoniae is an important drug target. In a study published in the Journal of Medicinal Chemistry in 2018 [5], a novel class of beta-lactamase inhibitors was successfully developed. These inhibitors exhibited exceptional inhibitory activity and demonstrated significant reversal of resistance in Gram-positive strains of Streptococcus pneumoniae. This new compound was engineered through structural optimization and drug design, resulting in tighter binding to the bacterium’s beta-lactamase enzyme, thus inhibiting its activity and restoring the efficacy of beta-lactam antibiotics for effective infection treatment.

Antibiotic research, particularly in the context of designing protein targets to combat bacterial resistance, has become a vibrant field. In this field, many potential drugs are actively under development [5].

Advancements in biotechnology and computer simulation methods have enabled researchers to better understand the molecular mechanisms of diseases and predict potential drug targets. This has allowed for more precise drug modifications that target specific molecular pathways, ultimately improving treatment efficacy while reducing the risk of adverse reactions [6].

Therefore, taking clindamycin derivatives as an example, we aimed to investigate the specific antimicrobial mechanisms underlying these derivatives. By focusing on clindamycin as a case study, we intended to explore the drug targets and mechanisms responsible for the broad-spectrum antimicrobial activity observed in clindamycin derivatives. This investigation provides new insights and theoretical foundations for the clinical pharmacological effects of clindamycin derivatives.

## 2. Results

### 2.1. Screening of Clindamycin Targets

A total of 300 clindamycin targets were screened through the Swiss Target Prediction database and ranked from high to low according to the likelihood of the targets. The targets are shown in Appendix A. The target classes of the three compounds are shown in Figure 1. The target number of Family A G-protein-coupled receptor ranked first, as shown in Figure 1A,B. The target numbers of kinase, enzyme and family A G-protein-coupled receptor ranked as the top three, as shown in Figure 1C.

### 2.2. Intersection of Three Clindamycin Derivative Targets

Through Venny 2.1.0 (csic.es) (https://bioinfogp.cnb.csic.es/tools/venny/index.html, accessed on 19 July 2023) online software, we screened 26 targets from the 300 targets of three compounds, as shown in Figure 2. The targets are shown in Table 2.

### 2.3. Screening of the Antibacterial Targets of the Clindamycin Derivatives through the PubChem Database

Using PubChem (nih.gov) (https://pubchem.ncbi.nlm.nih.gov/, accessed on 19 July 2023), after analyzing the target organisms of the 26 intersecting targets, we filtered out six protein targets. Using RCSB PDB (https://www.rcsb.org/, accessed on 20 July 2023), based on the antimicrobial profile of the clindamycin derivatives, we screened a number of protein targets, as shown in Table 3.

### 2.4. Molecular Docking Simulation and Validation

We conducted a molecular docking study using Discovery Studio 2019 Client to investigate the binding interactions of a specific antibiotic compound, denoted as compound **3**. In this study, we considered five distinct protein targets as potential binding partners for compound **3**. To assess the quality of the docking results, we employed the LibDock score, a scoring system that quantifies the binding affinity between molecules. Docking results were considered significant if they achieved a LibDock score exceeding 140, indicating a substantial binding affinity, as shown in Table 4. Ultimately, we identified and summarized five protein targets with docking scores above this threshold. These targets represent potential candidates for the interaction with and modulation of compound **3**.

### 2.5. Stability of the Docked Complexes Studied via MD Simulation

We conducted further conformational screening of the compound using molecular dynamics simulations. Specifically, we performed MD simulations involving multiple targets from five proteins along with compound **3**. Throughout these simulations, we monitored the root mean square deviation (RMSD) values of the entire system. As illustrated in Figure 3, the RMSD values exhibited a gradual convergence during the course of the simulations and eventually reached a stable state. Based on the RMSD results, we identified and selected four protein targets that demonstrated stable and reasonable conformational behavior.

In summary, our study utilized molecular dynamics simulations to assess the conformational stability of compound **3** in the presence of multiple target proteins. The convergence and stability of RMSD values guided our selection of the four protein targets for further investigation.

### 2.6. Binding Force Analysis

In light of the results of the MD simulations, we performed a binding force analysis using Discovery Studio’s receptor–ligand interaction calculation tool in Figure 4. The interactions between protein (PDB ID: 8c5p) and compound **3** mainly comprised hydrogen bonds, and electrostatic and hydrophobic interactions in Figure 4A. For the hydrogen bonds, residue 117 and residue 41 formed two conventional hydrogen bonds with bond lengths of 2.4 Å and 3.08 Å. Residue 71, residue 40, and residue 164 formed two carbon–hydrogen bonds. For the electrostatic interactions, residue 124 formed one pi–cation bond. For the hydrophobic interactions, residues 38, 39, and 145 formed one pi–alkyl bond, two amide–pi-stacked bonds, and one alkyl bond. The interactions between protein (PDB ID: 7p2x) and compound **3** mainly comprised hydrogen bonds, and electrostatic, hydrophobic, and miscellaneous interactions in Figure 4B. For the hydrogen bonds, residue 77 and residue 76 formed two conventional hydrogen bonds with bond lengths of 2.64 Å and 3.10 Å. Residue 46, residue 49, and residue 77 formed four carbon–hydrogen bonds. For the electrostatic interactions, residue 124 formed one pi–cation bond. For the hydrophobic interactions, residues 16, 43, 47, 71, 78, 94, 120, and 165 formed five pi–alkyl bonds, two amide–pi stacked bonds, one pi–sigma bond, and two alkyl bonds. The interactions between protein (PDB ID: 4qy5) and compound **3** mainly comprised hydrogen bonds, and electrostatic, and hydrophobic interactions in Figure 4C. For the hydrogen bonds, residue 281 and residue 277 formed two conventional hydrogen bonds with bond lengths of 2.82 Å and 3.02 Å. Residue 219, residue 220, and residue 280 formed four carbon–hydrogen bonds. For the electrostatic interactions, residue 32 and residue 288 formed four pi–cation bonds. For the hydrophobic interactions, residues 221, 225, 287, and 287 formed four alkyl bonds, and one pi–alkyl bond. The interactions between protein (PDB ID: 7ohn) and compound **3** mainly comprised hydrogen bonds, and electrostatic, hydrophobic, and miscellaneous interactions in Figure 4D. For the hydrogen bonds, residue 166 formed two conventional hydrogen bonds with a bond length of 2.26 Å and 2.38 Å. Residue 166 and residue 105 formed four carbon–hydrogen bonds. Residue 166 and residue 105 formed three pi–donor hydrogen bonds. For the electrostatic interactions, residue 32 and residue 288 formed two pi–cation bonds. For the hydrophobic interactions, residue 179 formed one alkyl bond, and residues 183, 179, and 71 formed four pi–alkyl bonds.

### 2.7. ADMET Prediction

We used the ADMET Descriptors module and the TOPKAT module in Discovery Studio software 2018 to predict the ADMET and toxicological properties of the compounds.

ADMET predictions indicated that compound **3** was soluble in water at 25 °C (log (SW) = −3.802). The ADMET_EXT_CYP2D6 was −2.93908, which did not inhibit cytochrome P4502D6. The ADMET_EXT_ Hepatoxic was −16.0559, which was the lowest toxicity to the liver. Plasma protein models showed that carrier proteins in the blood do not affect the efficiency of the drug (ADMET_EXT_PPB = −5.27966).

Toxicity predictions were calculated using the Ames heteroaromatic model. The computed rat oral lethal dose 50 (LD50) value for compound **3** obtained with the Rat Oral LD50 heteroaromatic model was 5525.31 mg/kg. These parameters were checked to be within their standard ranges, showing compound **3** is suitable for further development as a lead compound.

### 2.8. Protein Subcellular Localization

Based on the results of molecular docking and MD simulations, we identified β-lactamase, FleQ, FtsZ, and aspC as the most significant protein targets in Table 5. Subsequently, we conducted subcellular localization analysis for these proteins based on their amino acid sequences. Our findings revealed that FtsZ is primarily located in the cytoplasm (51.35%), but it is also present in the cytoplasmic nucleus, suggesting potential interactions in both compartments. β-lactamase primarily resides in the cytoplasm (38.36%), but it is also found in the cytoplasmic nucleus and cytoskeleton, indicating diverse interactions. FleQ is predominantly localized in the cytoplasm (31.63%), with a presence in the cytoplasmic nucleus and cytomitome, implying roles in these compartments. aspC is mainly in the cytoplasm (59.26%). Therefore, these four protein targets play roles in multiple cellular compartments within bacteria. 

## 3. Materials and Methods

The experimental procedure was performed with an Intel^®^ Xeon^®^ CPU E5-2650 0 @2.00 GHz (Intel, Santa Clara, CA, USA) processor, using a Windows 10 (Microsoft Corporation, Redmond, WA, USA) operating system and a 4 GB NVIDIA Quadro 2000 graphics card (Nvidia, Santa Clara, CA, USA). vmd1.9.3 was used as a 3D visualization window. 

### 3.1. Screening of Clindamycin Targets

In our previous research, we observed that compounds **3**, **3e**, and **4** demonstrated broad-spectrum antibacterial activity against four strains of Gram-negative bacteria (*Pseudomonas aeruginosa*, *Klebsiella pneumoniae*, *Salmonella* spp., and *Escherichia coli*), one strain of Gram-positive bacteria (*Staphylococcus aureus*), and a fungal strain (*Candida albicans*). Therefore, in light of this characteristic, the present study employed the Swiss Target Prediction database (http://swisstargetprediction.ch, accessed on 19 July 2023) using the following chemical structures as input, respectively [7]: SMILES code of compound **3**, compound **4**, and compound **3e**. These compounds were used as keywords to predict potential drug targets for the three clarithromycin derivatives. The species Homo sapiens was selected for target prediction, leading to the identification of potential therapeutic targets for these compounds.

### 3.2. Intersection of Three Clindamycin-Derivative Targets

Based on the shared broad-spectrum antibacterial activity against bacteria and fungi exhibited by designed compounds **3**, **3e**, and **4**, we employed the online tool “https://bioinfogp.cnb.csic.es/tools/venny/index.html, accessed on 19 July 2023” to intersect the 100 target proteins predicted for compound **3**, the 100 target proteins predicted for compound **3e**, and the 100 target proteins predicted for compound **4** [8]. This intersection resulted in a set of common target proteins.

### 3.3. Screening of Antibacterial Targets of Clindamycin Derivatives through PubChem Database

In order to further screen the targets and study the interactions between the targets, we screened the targets of 26 proteins through the PubChem database (https://pubchem.ncbi.nlm.nih.gov/, accessed on 20 July 2023) [9], and the keywords “bacteria, fungi, *Staphylococcus aureus*, *Escherichia coli*, *Pseudomonas aeruginosa*, and *Candida albicans*” were used to screen out the protein targets of the drug.

### 3.4. Molecular Docking Simulation and Validation

In this process, we created a ligand library using Discovery Studio 2019 Client, performed docking with CHARMM to refine ligand shapes and charge distribution, and analyzed binding interactions between clindamycin derivatives and drug targets. We selected the best poses based on LibDock scores, filtering targets with scores over 140, providing valuable insights into the binding mechanisms for our research [10].

### 3.5. Stability of the Docked Complexes Studied via MD Simulation

Molecular dynamics (MD) simulations using GROMACS (version 2020.3) were used to analyze protein–ligand binding dynamics [11]. Protein structures were optimized with the AMBER99SB-ILDN force field [12], and water molecules were modeled with the TIP3P model [13,14]. ACPYPE was used to calculate the ligand charges and generate GAFF force-field-compatible files [15]. Simulations employed cubic boxes with a minimum atom-box boundary distance of 0.8 nm, hydrated with SOL water at a 1000 g/L density. Chloride ions replaced solvent water for electrical neutrality. An initial energy minimization step relaxed the system, followed by a 100 ps restrained MD simulation at 298.15 K. Unrestricted MD simulations with a time step of 0.002 ps were performed for 10 ns and 30 ns, maintaining isothermal-isobaric conditions at 298.15 K and 1 bar pressure, controlled using thermostats and barostats [16].

### 3.6. Calculation of the Binding Energy

The binding free energy between the protein–ligand complexes was estimated using the MM/PBSA equation [17]. The APBS lattice parameters are output according to the MD results, and APBS module using Discovery Studio 2019 Client (Pacific Northwest National Laboratory, Richland, DC, USA) [18] was applied to calculate polar solvation energy (PB) and nonpolar solvation energy (SA). The binding free energy (Δ*G*_*Bind*) is given by the following equation: 


Δ*G*_*Bind* = Δ*G*_*Complex* − Δ*G*_*Ligand* − Δ*G*_*Receptor*


The MM energy accounts for the intermolecular interactions (e.g., van der Waals forces, electrostatic interactions, and hydrogen bonding) between the protein and the ligand. It is typically calculated using a force field that approximates the potential energy function of the system. Polar solvation energy (Δ*G*_PB) describes the interaction between solvent molecules (usually water molecules) and polar atoms (partially charged) in proteins and ligands. Nonpolar solvation energy (Δ*G*_SA) describes the interaction between solvent molecules and nonpolar regions (usually hydrophobic hydrocarbon chains) in proteins and ligands.

### 3.7. ADMET Prediction

We used the ADMET Descriptors and Toxicity Prediction modules in Discovery Studio software 2018 to predict the ADMET and toxicological properties of compounds [19].

The ADMET Descriptors module was developed based on descriptors computed by linear formulas. The water-soluble model was developed using a training set of 775 compounds and a test set of 34 compounds using descriptors of solubility and solubility levels (R^2^ = 0.88, SD = 0.79).

The TOPKAT module was used to analyze the structure of a compound, identify functional groups and substructures, convert them into numerical descriptors, and compare them to known compounds to predict potential toxicity based on structural relationships and quantitative models.

### 3.8. Protein Subcellular Localization

Subcellular localization refers to the precise cellular location of a protein or its gene expression product, encompassing compartments such as the nucleus, cytoplasm, and cell membrane. This spatial organization is crucial for the protein’s proper function, as it ensures access to the necessary chemical environment and interacting factors. Misplacement can disrupt cellular processes, making understanding protein localization essential for studying gene functions, protein interactions, and their mechanisms [20]. 

PSORT is a computer program used to predict the subcellular location of proteins based on their amino acid sequences and source information. To predict subcellular localization, the amino acid sequences of target proteins are obtained from the UniProt database, and these sequences are then inputted into the PSORT II online software (https://psort.hgc.jp/, accessed on 21 July 2023). PSORT II provides predictions regarding the subcellular location of these target proteins [21].

## 4. Conclusions

In the course of our prior research, we made a significant revelation: clindamycin derivatives possess an exceptional broad-spectrum antibacterial activity, effectively targeting Gram-positive bacteria, Gram-negative bacteria, and fungi. Building upon this discovery, our study aimed to delve deeper into the molecular mechanisms underlying this remarkable property and identify potential antibacterial targets.

Our initial step involved the utilization of the Swiss Target Prediction database to explore the vast landscape of target proteins among the 300 clindamycin derivatives. Interestingly, our analysis revealed that kinase, enzyme, and family A G-protein-coupled receptor were the top three most abundant protein classes among these derivatives. Given the structural similarities of these clindamycin derivatives, we employed Venny 2.1.0 (csic.es) to identify 26 overlapping target proteins. Subsequently, guided by the PubChem database, we strategically filtered these targets based on their relevance to antibacterial activities, encompassing pathogens like *Staphylococcus aureus*, *Pseudomonas aeruginosa*, *Candida albicans*, and *Escherichia coli*. Our rigorous screening process yielded a select set of six antibacterial target proteins, consisting of five family A G-protein-coupled receptors and one kinase. To determine the binding sites for these compounds, we employed compound **3** as a representative molecule and conducted molecular docking simulations with these six target proteins. From these simulations, we identified five protein targets with LibDock scores exceeding 140, indicating a substantial binding affinity. For further validation, we subjected these interactions to extensive molecular dynamics simulations, spanning durations of 10 and 30 nanoseconds. Analysis of the root mean square deviation (RMSD) results narrowed our selection down to four protein targets, namely β-lactamase, FleQ, FtsZ, and aspC, each demonstrating stable and biologically plausible conformational behavior.

Analyzing the binding forces, we observed that compound **3** formed hydrogen bonds, electrostatic interactions, and hydrophobic interactions with these four target proteins. Particularly noteworthy was the favorable electrostatic interaction between compound **3**’s aromatic structure and the target proteins. To assess the safety and cellular activity of compound **3**, we conducted ADMET predictions, reassuringly finding that compound **3** exhibits a safe cellular profile. Furthermore, to gain insights into the cellular locations where clindamycin derivatives exert their antibacterial effects, we performed protein subcellular localization for the four identified target proteins. The results were enlightening, revealing that FtsZ and aspC predominantly reside in the cytoplasm, β-lactamase primarily localizes in cyto_nucl and cysk, and FleQ is primarily distributed across cyto, cyto_nucl, and cyto_mito compartments.

In summary, our comprehensive investigations shed light on the intricate mechanisms underpinning the broad-spectrum antibacterial activity of clindamycin derivatives. These findings not only hold promise for the development of innovative antimicrobial agents, but also open new horizons in the ongoing battle against antibiotic resistance. While we acknowledge the computational nature of our approach and the need for future experimental validation, our research sets the stage for potentially transformative clinical applications.

## Figures and Tables

**Figure 1 molecules-28-07357-f001:**
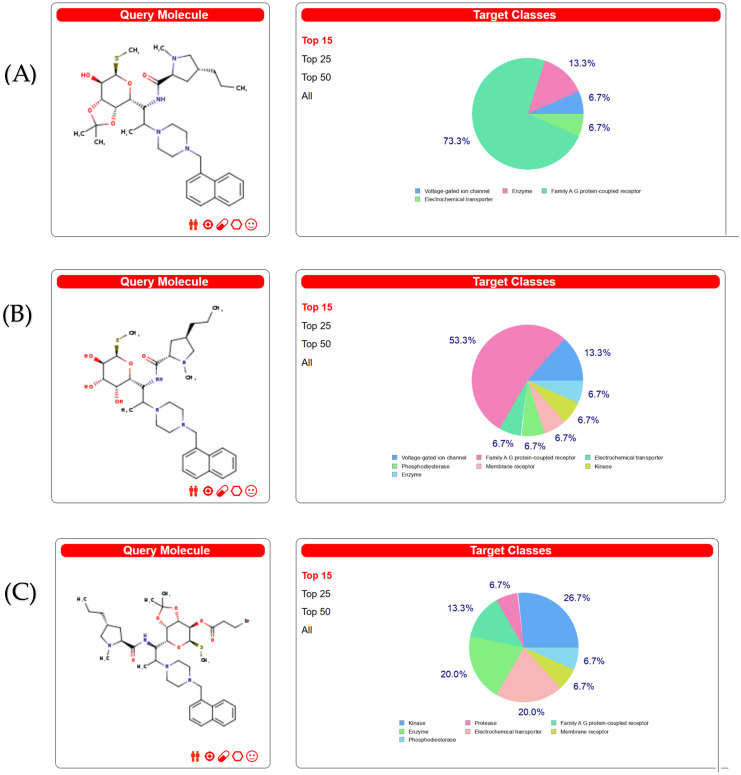
Target classes for three compounds. (**A**) Targets classes of compound **3**. (**B**) Targets classes of compound **3e**. (**C**) Targets classes of compound **4**.

**Figure 2 molecules-28-07357-f002:**
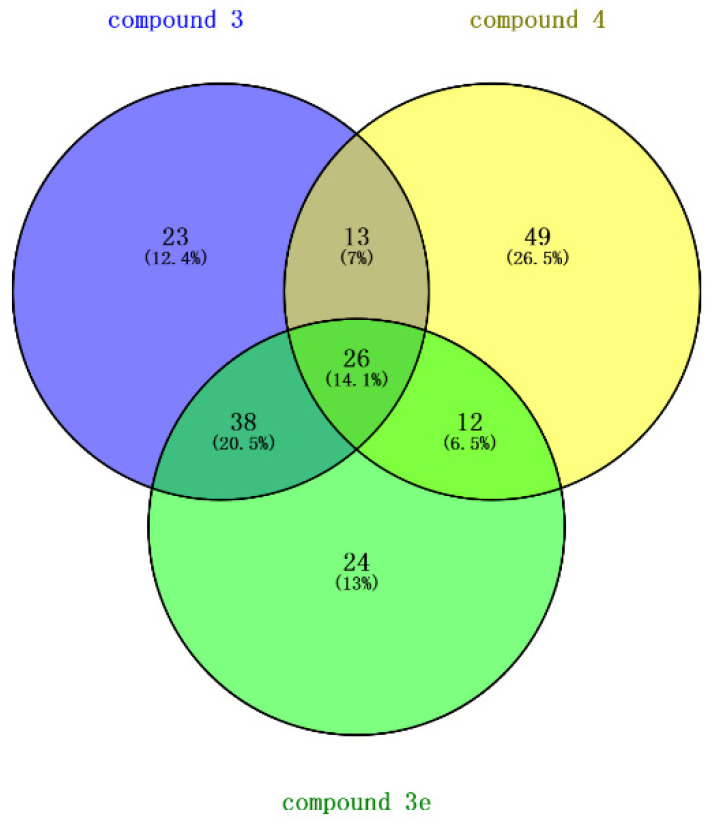
Venn diagram of targets from three compounds.

**Figure 3 molecules-28-07357-f003:**
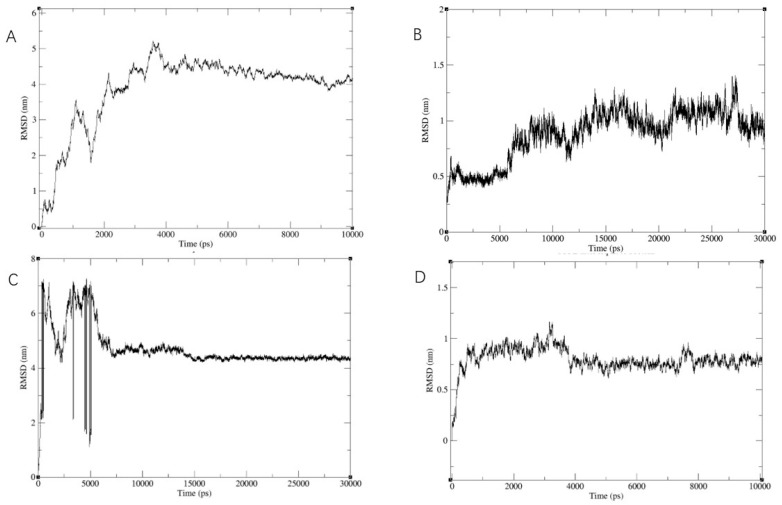
Results of the molecular dynamic simulations for four protein–compound **3** complexes. (**A**) RMSD of compound **3** with protein target (PDB ID: 8c5p) in 10 ns. (**B**) RMSD of compound **3** with protein target (PDB ID: 7p2x) in 30 ns. (**C**) RMSD of compound **3** with protein target (PDB ID: 4qy5) in 30 ns. (**D**) RMSD of compound **3** with protein target (PDB ID: 7ohn) in 10 ns.

**Figure 4 molecules-28-07357-f004:**
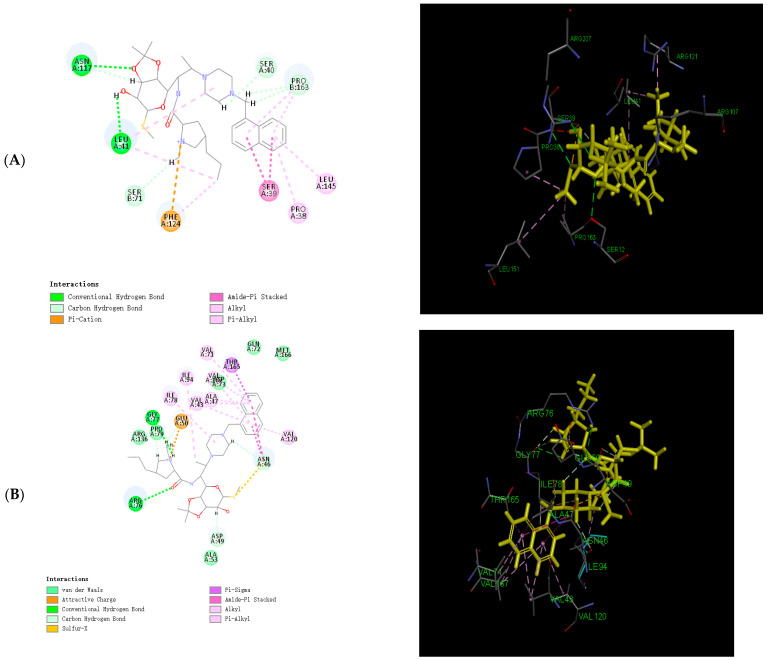
The 2D and 3D diagrams of results of docking analysis: (**A**) 2D and 3D diagrams of compound **3** with protein target (PDB ID: 8c5p); (**B**) 2D and 3D diagrams of compound **3** with protein target (PDB ID: 7p2x); (**C**) 2D and 3D diagrams of compound **3** with protein target (PDB ID: 4qy5); (**D**) 2D and 3D diagrams of compound **3** with protein target (PDB ID: 7ohn).

**Table 1 molecules-28-07357-t001:** Structures and antibacterial spectra of three compounds.

Compounds	Structural Formula	Antibacterial Spectrum
**3**	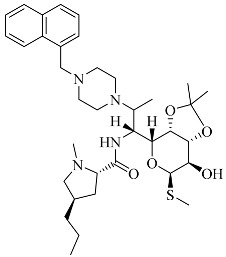	*E. coli* ATCC35218 *K. pneumoniae* ATCC700607 *S. enteritidis* CICC21482 *P. aeruginosa* ATCC27853 MRSA clinical isolate *C. albicans* CMCC98001
**3e**	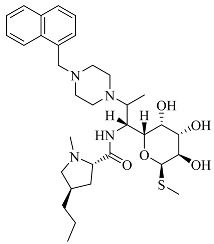	
**4**	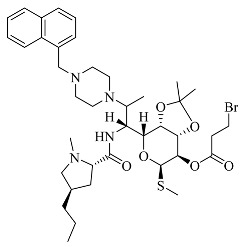	

**Table 2 molecules-28-07357-t002:** The screened targets using Venny.

Common Name	
Public Protein	Target
KCNH2	HERG
PIK3CA	PI3-kinase p110-alpha subunit
ADRA1D	Alpha-1d adrenergic receptor
ADRA1A	Alpha-1a adrenergic receptor
ADRA1B	Alpha-1b adrenergic receptor
PIK3CG	PI3-kinase p110-gamma subunit
HTR1A	Serotonin 1a (5-HT1a) receptor
SLC6A4	Serotonin transporter
CCR3	C-C chemokine receptor type 3
HTR2A	Serotonin 2a (5-HT2a) receptor
ADORA2A	Adenosine A2a receptor
F10	Thrombin and coagulation factor X
PIK3CB	PI3-kinase p110-beta subunit
MAPK14	MAP kinase p38 alpha
AURKA	Serine/threonine-protein kinase AuroraA
PDE5A	Phosphodiesterase
PDE4B	Phosphodiesterase 4B
PRKCB	Protein kinase C beta
SIGMAR1	Sigma opioid receptor
ADRB2	Adrenergic receptor beta
ADRB1	Beta-1 adrenergic receptor
CFD	Complement factor D
AURKB	Serine/threonine-protein kinase Aurora-B
ADORA1	Adenosine A1 receptor
RPS6KB1	Ribosomal protein S6 kinase 1
PIM1	Serine/threonine-protein kinase PIM1

**Table 3 molecules-28-07357-t003:** Results of screened targets using PubChem database.

Target	PDB ID	Uniprot ID	ChEMBL ID	Target Class
Adrenergic receptor beta	4QY5, 4QY6, 4ID4, 4R4R, 4R4S	P07550	CHEMBL210	Family A G-protein-coupled receptor
Beta-1 adrenergic receptor (by homology)	6H7J, 6H7L, 6H7M, 6H7N, 6H7O, 6IBL	P08588	CHEMBL213	Family A G-protein-coupled receptor
Adenosine A1 receptor	6D9H, 7LD4, 7LD3	P30542	CHEMBL226	Family A G-protein-coupled receptor
Mu opioid receptor (by homology)	8F7R	P35372	CHEMBL233	Family A G-protein-coupled receptor
Adenosine A2a receptor	5K2A, 5K2B,5K2D	P29274	CHEMBL251	Family A G-protein-coupled receptor
MAP kinase ERK2	8CJ0, 8C5P, 8C5F, 8C5E, 8BN6, 8BFT, 8BFR, 8BCJ, 8BCI, 7ZD3, 7ZD1, 7ZD0, 7Z2W, 7Z2T, 7Z18, 7Z15, 7VTG, 7VF8, 7VA3, 7V9Z, 7V6T, 7RZK, 7RM7, 7R2O, 7R1J, 7R1H, 7R0F, 7PTF, 7PJI, 7P8X, 7P8U, 7P2X, 7P2W, 7P2N, 7P2M, 7ONY, 7ON4, 7ON2, 7OJD, 7OJC, 7OJB, 7OI2, 7OHN, 7OHL, 7OHK, 7OHH, 7O5M, 7KYE, 7KRV, 7KRU, 7KPS, 7KPP, 7KDR, 7KDO	P28482	CHEMBL4040	Kinase

**Table 4 molecules-28-07357-t004:** Results of docking analysis.

PDB ID	Absolute Energy	Clean Energy	Relative Energy	Lib Dock Score	Hot Spots (Average)
7PTF	76.0364	121.839	6.14328	142.136	16.65, −11.84, 50.63, A, 84, 29
					14.65, −7.44, 48.43, A, 58, 45
					16.05, −6.24, 46.63, A, 35, 46
7P2X	83.8745	121.839	13.9814	143.521	−22.65, −17.99, 6.92, A, 15, 21
					−16.45, −11.99, 8.72, A, 57, 35
					−19.85, −10.79, 9.52, A, 67, 43
7OHN	84.5974	121.839	14.6402	162.369	0.48, −5.97, 21.08, A, 65, 26
					4.68, −11.38, 25.28, A, 83, 35
					6.08, −10.57, 27.28, A, 98, 37
7O5M	84.6001	121.839	14.7069	166.496	17.98, 35.34, 18.83, A, 9, 25
					16.98, 36.54, 19.03, A, 13, 26
					9.58, 34.54, 33.63, A, 85, 41
8C5P	81.269	121.839	10.961	181.861	29.18, 13.58, 20.57, P, 45, 23
					35.38, 11.38, 14.77, A, 15, 35
					35.98, 11.78, 13.57, A, 8, 36

**Table 5 molecules-28-07357-t005:** Protein subcellular localization results.

Drug	Protein	Location (k = 23)	UniProt ID	PDB ID
Clindamycin derivatives	FtsZ	cyto: 51.35% cyto_nucl: 29.73% cysk: 10.81% pero: 5.40% mito: 2.70%:	Q2FZ89	7ohn
Clindamycin derivatives	β-lactamase	cyto: 38.36%, cyto_nucl: 26.03%, cysk: 19.18%, nucl: 8.22%, mito: 5.48%, pero: 2.74%	UPI00067E6531	4qy5
Clindamycin derivatives	FleQ	cyto: 31.63% cyto_nucl: 25.17%, cyto_mito: 19.73%, nucl: 14.29%, mito: 5.10%, cysk: 4.08%	UPI0021CDE869	7ptf
Clindamycin derivatives	aspC	cyto: 59.26%, nucl: 18.52%, mito: 14.81%, cysk: 7.41%	UPI000274A8A1	7p2x

## Data Availability

The data for the compound are provided in Appendix A.

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
