# Peer review of "Structure–Activity Relationship Target Prediction Studies of Clindamycin Derivatives with Broad-Spectrum Bacteriostatic Antibacterial Properties"

_molecules, 2023, doi:10.3390/molecules28217357_

Round 1

Reviewer 1 Report

The study under review explores the potential of clindamycin derivatives with broad-spectrum antibacterial properties. The authors employ molecular docking and dynamics simulations to investigate the antibacterial capabilities of three clindamycin derivatives against 300 target proteins.

The introduction provides a clear rationale for the study, highlighting the pressing issue of bacterial resistance and the need for novel antimicrobial agents. It adequately sets the context and justifies the investigation of clindamycin derivatives as a potential solution. However, it could benefit from a more detailed review of the existing literature on the subject.

The methodology section is well-structured and explains the experimental design clearly. The use of molecular docking and dynamics simulations is appropriate for the study's objectives. However, it would be beneficial to include more information about the software and parameters used for these simulations.

The presentation of results is generally clear and concise.

The discussion section should provide a more in-depth analysis of the results. It would be helpful to discuss how the identified protein targets relate to clindamycin derivatives' antibacterial properties and their potential in addressing bacterial resistance. The discussion should also consider the limitations of the study, such as the assumptions made in the molecular docking and dynamics simulations.

This study makes a valuable contribution to the field of antimicrobial research by exploring the potential of clindamycin derivatives as antibacterial agents. The use of molecular docking and dynamics simulations is a suitable approach for this investigation. However, there is room for improvement in the presentation of results and the depth of analysis in the discussion section. Additionally, providing more context in the introduction and a more comprehensive conclusion would enhance the study's overall impact.

Recomendation

  • Provide a more extensive literature review in the introduction to establish the study's relevance.
  • Offer more details about the software and parameters used in the molecular docking and dynamics simulations.
  • Discuss the implications of the identified protein targets in greater depth, including their potential in addressing bacterial resistance.
  • Acknowledge and discuss the limitations of the study.
  • Strengthen the conclusion by summarizing the study's contributions and suggesting avenues for future research.

acceptable

Author Response

Response to Reviewer 1 Comments

1. Summary

Thank you very much for taking the time to review this manuscript. Please find the detailed responses below and the corresponding revisions in the re-submitted files.

2. Questions for General Evaluation

Reviewer’s Evaluation

Response and Revisions

Does the introduction provide sufficient background and include all relevant references?

Yes

Thanks for the reply, I've made a revision in yellow in the re-submitted file.

Are all the cited references relevant to the research?

Yes

Is the research design appropriate?

Yes

Are the methods adequately described?

Yes

Are the results clearly presented?

Yes

Are the conclusions supported by the results?

Yes

3. Point-by-point response to Comments and Suggestions for Authors

Comments 1:

1. The study under review explores the potential of clindamycin derivatives with broad-spectrum antibacterial properties. The authors employ molecular docking and dynamics simulations to investigate the antibacterial capabilities of three clindamycin derivatives against 300 target proteins.

2. The introduction provides a clear rationale for the study, highlighting the pressing issue of bacterial resistance and the need for novel antimicrobial agents. It adequately sets the context and justifies the investigation of clindamycin derivatives as a potential solution. However, it could benefit from a more detailed review of the existing literature on the subject.

3. The methodology section is well-structured and explains the experimental design clearly. The use of molecular docking and dynamics simulations is appropriate for the study's objectives. However, it would be beneficial to include more information about the software and parameters used for these simulations.

The presentation of results is generally clear and concise.

4. The discussion section should provide a more in-depth analysis of the results. It would be helpful to discuss how the identified protein targets relate to clindamycin derivatives' antibacterial properties and their potential in addressing bacterial resistance. The discussion should also consider the limitations of the study, such as the assumptions made in the molecular docking and dynamics simulations.

5. This study makes a valuable contribution to the field of antimicrobial research by exploring the potential of clindamycin derivatives as antibacterial agents. The use of molecular docking and dynamics simulations is a suitable approach for this investigation. However, there is room for improvement in the presentation of results and the depth of analysis in the discussion section. Additionally, providing more context in the introduction and a more comprehensive conclusion would enhance the study's overall impact.

6. Recomendation

Provide a more extensive literature review in the introduction to establish the study's relevance.

Offer more details about the software and parameters used in the molecular docking and dynamics simulations.

Discuss the implications of the identified protein targets in greater depth, including their potential in addressing bacterial resistance.

Acknowledge and discuss the limitations of the study.

Strengthen the conclusion by summarizing the study's contributions and suggesting avenues for future research.

Response 1: Thank you for your careful review and for your valuable comments.

Response 2: Thank you for your careful review and for your valuable comments. In lines 23-29 of the revised manuscript, we have added some content to make the content more complete.

Response 3: Thank you for your careful review and for your valuable comments. We have presented the results required for molecular docking and molecular dynamics simulations in the results. If you need any more information, please provide detailed examples and we will provide you with more relevant information.

Response 4: Thank you for your careful review and for your valuable comments. We have revised the conclusions of manuscript in 296 - 317.

Reviewer 2 Report

Please be careful with non-optimal ley-out features such as:

lines 100-6: no need for the SMILES code in the text

-line 147: "The [...] equation is given by thew following equation:" (and there appears to be no equation after this)

-line 162: "R2" (2 should be written in superscript")

-line 181: "Screening of Selemetinib targets" (should be in italics since it's a subheading)

-line 184: Table 2 is very difficult to follow, I'd add it as a Supplementary Table

-line 199: "Figure 2 Venn of targets from three compounds" (please check size and font)

-lines 232-259: check odd spaces from that paragraph

-lines 261-4: check line spacing

-lines 297-317: Conclusions section have smaller size than the rest of the text

Also, I'd add the information from Supplementary Table 1 to Table 1 in the text, and include the chemical experimental part in the main text too.

Please be careful with typos and wrong verbal forms such as:

- line 121: "and the source is The keywords"

-line 184: "Tabble 2" (should be Table 2)

-lines 195-6: "we screens 26 targets" (should change the word screens by screened); "The targets is shown" (should change is by are)

-line 187: "The target numbers [...] is the top three in Figure 1-C" (should substitute the word is by are)

-lines 202-4: "we filters (should change filters by filtered); ", Based on" (why the capital letter after a comma?); "as shown in the Table 4" (should be written as shown in Table 4)

Author Response

Response to Reviewer 2 Comments

1. Summary

Thank you very much for taking the time to review this manuscript. Please find the detailed responses below and the corresponding revisions in the re-submitted files.

2. Questions for General Evaluation

Reviewer’s Evaluation

Response and Revisions

Does the introduction provide sufficient background and include all relevant references?

Can be improved

Thanks for the reply, I've made a revision in blue in the re-submitted file.

Are all the cited references relevant to the research?

Can be improved

Is the research design appropriate?

Can be improved

Are the methods adequately described?

Can be improved

Are the results clearly presented?

Can be improved

Are the conclusions supported by the results?

Can be improved

3. Point-by-point response to Comments and Suggestions for Authors

Comments 1:

1. Comments and Suggestions for Authors

Please be careful with non-optimal ley-out features such as:

lines 100-6: no need for the SMILES code in the text

-line 147: "The [...] equation is given by thew following equation:" (and there appears to be no equation after this)

-line 162: "R2" (2 should be written in superscript")

-line 181: "Screening of Selemetinib targets" (should be in italics since it's a subheading)

-line 184: Table 2 is very difficult to follow, I'd add it as a Supplementary Table

-line 199: "Figure 2 Venn of targets from three compounds" (please check size and font)

-lines 232-259: check odd spaces from that paragraph

-lines 261-4: check line spacing

-lines 297-317: Conclusions section have smaller size than the rest of the text

Also, I'd add the information from Supplementary Table 1 to Table 1 in the text, and include the chemical experimental part in the main text too.

Comments on the Quality of English Language

Please be careful with typos and wrong verbal forms such as:

- line 121: "and the source is The keywords"

-line 184: "Tabble 2" (should be Table 2)

-lines 195-6: "we screens 26 targets" (should change the word screens by screened); "The targets is shown" (should change is by are)

-line 187: "The target numbers [...] is the top three in Figure 1-C" (should substitute the word is by are)

-lines 202-4: "we filters (should change filters by filtered); ", Based on" (why the capital letter after a comma?); "as shown in the Table 4" (should be written as shown in Table 4)

Response 1: Thank you for your careful review and for your valuable comments. For non-optimal ley-out features, we have revised the paper in blue.

Response 2: Thank you for your careful review and for your valuable comments. “Also, I'd add the information from Supplementary Table 1 to Table 1 in the text, and include the chemical experimental part in the main text too.” Sorry, the data related to chemical experiments cannot be provided, and the chemical experimental data in the supporting information cannot be published in this journal. Because this experimental part belongs to previous research.

Response 3: Thank you for your careful review and for your valuable comments. We have reversed typos and wrong verbal forms in blue.

Round 2

Reviewer 2 Report

The manuscript has been duly corrected according to the suggestions